# Analytical Method for Measurement of Tobacco-Specific Nitrosamines in E-Cigarette Liquid and Aerosol

**Yoon-Seo Lee [1], Ki-Hyun Kim [1,*], Sang Soo Lee [2], Richard J. C. Brown [3] and Sang-Hee Jo [1]**

[1] Department of Civil and Environmental Engineering, Hanyang University, 222 Wangsimni-ro, Seoul 04763, Korea; dbstj_j@naver.com (Y.-S.L.); sangnimlove@naver.com (S.-H.J.)

[2] Department of Environmental Engineering, Yonsei University, Wonju 26493, Korea; cons@yonsei.ac.kr

[3] Department of Chemical, Medical and Environmental Science, National Physical Laboratory, Teddington TW11 0LW, UK; richard.brown@npl.co.uk

* Correspondence: kkim61@hanyang.ac.kr; Tel.: +82-2-2220-2325; Fax: +82-2-2220-0399

**Abstract:** An experimental method was developed and validated for the collection and analysis of tobacco-specific nitrosamines (TSNAs) that are present in electronic cigarette (EC) liquid or are released from aerosol samples using a liquid chromatography-tandem mass spectrometry (LC-MS/MS) system. As part of this study, the relative recovery of four target TSNAs was assessed by spiking standards in a mixture of propylene glycol and vegetable glycerin. Recovery was assessed against two major variables: (1) the chemical media (solution) selected for sample dilution (acetonitrile [ACN] vs. ammonium acetate [AA]) and (2) the type of sampling filter used (Cambridge filter pad [CFP] vs. quartz wool [QW] tube). The average recovery of TSNAs in terms of variable 1 was $134 \pm 22.1\%$ for ACN and $92.6 \pm 8.27\%$ for AA. The average recovery in terms of variable 2 was $83.4 \pm 7.33\%$ for QW and $58.5 \pm 12.9\%$ for CFP. Based on these conditions, the detection limits of $N'$-nitrosonornicotine (NNN), 4-(methylnitrosamino)-1-(3-pyridyl)-1-butanone (NNK), $N'$-nitrosoanatabine (NAT), and $N'$-nitrosoanabasine (NAB) were calculated as 4.40, 4.47, 3.71, and 3.28 ng mL$^{-1}$, respectively. The concentration of TSNAs in liquid and aerosol samples of six commercial EC solutions was measured as below the detection limit.

**Keywords:** tobacco specific-nitrosamines; electronic cigarette; liquid chromatography-tandem mass spectrometry

## 1. Introduction

An electronic cigarette (EC) is an electronic spraying device that generates aerosols containing nicotine. Unlike a conventional cigarette, an EC consists of a rechargeable battery, a cartridge containing the refill solution and aerosol, and an atomizer that transforms the liquid into an aerosol by heating the liquid solution [1–3]. The EC liquid is made up mainly of propylene glycol (PG) and/or vegetable glycerin (VG), to which flavors and/or nicotine can be added according to the consumer's preference. The use of ECs is highly popular around the world due to its ease, pleasant odors, and inhalation characteristics that are similar to conventional cigarettes [4,5]. In spite of a rapid increase in demand for EC, the content of potentially harmful compounds in the liquids before and after vaporization is unclear. Consequently, some countries are regulating the use of ECs very strictly [4].

It is reported that up to 7000 chemicals are present in a conventional cigarette and its smoke [6,7]. According to the International Agency for Research on Cancer (IARC), tobacco smoke contains 60 carcinogens, including aldehydes (e.g., formaldehyde and acetaldehyde), heavy metals (e.g., Cr),

nitrosamines (e.g., $N'$-nitrosonornicotine [NNN], and 4-(methylnitrosamino)-1-(3-pyridyl)-1-butanone [NNK]), polycyclic aromatic hydrocarbons (PAHs, benzeo[a]pyrene), and aromatic amines [8–10]. Moreover, there are more than 300 nitrosamines, among which tobacco-specific nitrosamines (TSNAs) have been identified as potent carcinogens. NNN and NNK are known to induce carcinogenesis through DNA adductions and mutations in addition to promoting tumor growth through receptor-mediated effects [11], and they are classified in the first (most dangerous) group of carcinogens [12]. $N'$-nitrosoanatabine (NAT) is classified in the third group of carcinogens. NAT is generated together with NNN and NNK in tobacco [11,13]. $N'$-nitrosoanabasine (NAB) is also classified in the third group of carcinogens, although its metabolic pathway has not yet been identified [14]. In addition, although 4-(methylnitrosamino)-4-(3-pyridyl)-butyric acid (iso-NNAC), 4-(methylnitrosamino)-4-(3-pyridyl)-1-butanol (iso-NNAL), and 4-(methylnitrosamino)-1-(3-pyridyl)-1-butanol (NNAL) were also reported in the monograph on Smokeless Tobacco [15], these three compounds have not been considered in the present evaluation due to the limited availability of toxicity data. Consequently, four TSNA components (NNN, NNK, NAT, and NAB) were selected as target components in this study.

TSNAs are generated by the nitrosation of nitrogen oxides ($NO_x$) and some alkaloids (the secondary amines nornicotine, anabasine, and anatabine and the tertiary amine nicortine) during air curing at 20 °C and pH 2–7 as part of the tobacco harvesting process [16–18]. Most of the nornicotine is made directly from nicotine through activity of the nicotine $N$-demethylating enzyme during the aging and curing process of mature leaves. The amount of TSNAs in mainstream cigarette smoke is directly related to compounds such as nitrate in hardened tobacco leaves [19–21]. Although the absolute amount of TSNAs is very low, the quantitative assessment of NNN and NNK was recommended to better understand the link between these compounds and human health effects because of the high frequency of lung cancer that is caused by smoking [22].

Levels of TSNAs in tobacco and mainstream cigarette smoke have been determined using various combinations of chromatographic analytical systems, such as a gas chromatography-thermal energy analysis (GC-TEA) [23], gas chromatography-mass spectrometry (GC/MS) [24], and liquid chromatography-tandem mass spectrometry (LC-MS/MS) [25–30]. The LC-MS/MS system was found to have significantly enhanced sensitivity and reproducibility relative to other instruments (e.g., superior detectability over GC/MS). Also, LC-MS/MS is capable of detecting NNN and NNK within a short analysis time [29,31–33]. To date, an analysis method for EC solution using an LC-MS/MS system has been proposed by a limited number of study groups (e.g., [34–37]). Moreover, there are only a few reported analytical methods for the measurement of TSNA concentration levels in EC aerosols. These methods captured the aerosol after the solvent had been injected into a gas-tight syringe or extracted the glass fiber filter pad that captured the aerosols with water [19,38]. As such, there are very limited case studies that evaluate the emissions of TSNAs released by the use of ECs.

In commercial EC solutions, various kinds of nicotine, such as synthetic nicotine, natural nicotine, and nicotine extracted from the stem are used [39]. The experiment that was performed in this study assumed that the target compounds in synthetic nicotine are same as nicotine in tobacco. A simple method was developed to quantify TSNAs from ECs in both solution and vaped aerosol samples (filter sampling) using a LC-MS/MS system. The EC aerosol was sampled through an automatic sampling device for EC smoke. The analysis conditions were established using comparable international standard methods (International Organization for Standardization (ISO), World Health Organization (WHO), and Cooperation Centre for Scientific Research Relative to Tobacco (CORESTA N° 75)) established for mainstream cigarette smoke, thereby maintaining consistency with previous methods that were used to obtain this type of data. To ensure the reliability of the analysis method, the relative recovery (RR) for four TSNAs was evaluated by spiking TSNA primary standards into the PG/VG mixed solution. Quantitative analysis of TSNAs was then carried out using six different commercial EC products in both solution and aerosol under the experimental conditions that were established

in this work. The results of our study will offer valuable insights into the development of reliable analytical methods for important components contained or generated due to the use of EC.

## 2. Materials and Methods

Two experiments were conducted to determine the dilution factor for the analysis of TSNAs in this study. As part of this study, the RRs of four target TSNAs were assessed by spiking standards in a PG/VG mixture. Performance against two major variables was assessed: (1) differences in the chemical media (solution) selected for sample dilution (ACN vs. AA (100 mmol)); and, (2) differences in the type of sampling filter used (Cambridge filter pad [CFP] vs. quartz wool [QW] tube). To assess the performance of these variables, the RRs of four TSNAs were evaluated in two stages of experiments.

### 2.1. Working Standard Solution for TSNAs

For the calibration analysis of TSNAs, primary standard (PS) solutions containing 1000 µg mL$^{-1}$ ($\pm$)-N′-nitrosonornicotine (NNN), 4-(methylnitrosamino)-1-(3-pyridyl)-1-butanone (NNK), (s)-N-nitrosoanabasine (NAB) dissolved in methanol, and 5 mg of N-nitrosoanatabine (NAT) powder were purchased (Certified Reference Materials for NNN, NNK, and NAB, Sigma-Aldrich Co., St. Louis, USA). First, NAT powder was dissolved in 99.8% methanol (Sigma-Aldrich Co.) to prepare a 5 µg mL$^{-1}$ solution (NAT-LS). Liquid working standards (L-WS) were then prepared by diluting three PS (NNN, NNK, and NAB) and NAT-LS with 99.9% acetonitrile (ACN, Sigma-Aldrich Co.) at seven concentrations (0.5, 1, 2, 5, 10, 50, and 100 ng mL$^{-1}$) in 2 mL vials. The basic physicochemical information of the four target TSNAs to be analyzed is shown in Supplementary Table S4 online. For a background sample (blank EC solution), a mixture of propylene glycol (PG, USP, Sigma-Aldrich Co.) and glycerin (VG, USP, Sigma-Aldrich Co.), which are the main components of EC, was prepared at a weight ratio of 1:1.

### 2.2. Pretreatment of the EC Solution

The dominant content (>80%) of EC solution is PG and VG. Flavor or nicotine may be added to this mixture according to the preference of the EC user [40]. Note that PG and VG have higher viscosity values than water (the usual matrix for LC analysis). Hence, if the EC sample is analyzed without a pretreatment step (such as dilution), it is difficult to do the LC-based analysis of TSNA [41–43].

To compare the required dilution to successfully elute TSNA in EC solution, two kinds of solvent, ammonium acetate (AA) and ACN, were selected. Previously, AA was used as the extraction solution for the analysis of TSNAs by ISO and WHO [16]. ACN was selected as an additional dilution solution to test in this study, because it can elute many chemicals in HPLC and it has similar performance to methanol. ACN is usually used as salting-out extraction solvent for analyzing N-nitrosamines [44].

In the first liquid experiment stage (L-S1), to determine the dilution factor, TSNA spiked samples were prepared at a concentration of 100 ng mL$^{-1}$ by adding three PSs and NAT-LS to a PG/VG mixture (1:1, $w/w$). The spiked sample was systematically diluted 20, 50, and 100 fold using each of the two dilution solvents (AA or ACN) and 1 µL of each diluted sample was analyzed by LC-MS/MS. Triplicate analysis for each sample was performed. In the second liquid experiment stage (L-S2), the spiked samples were prepared at three concentrations of 50, 250, and 500 ng mL$^{-1}$ by adding three PSs and NAT-LS into the PG/VG mixture. These spiked samples were then diluted with AA or ACN at a fixed dilution factor of 50, which was determined by the result of L-S1 analysis. One microliter of each diluted sample was analyzed by a LC-MS/MS. Triplicate analysis for each sample was conducted.

### 2.3. Aerosol Sample and Pretreatment Method for EC Aerosol

For analysis of the aerosol generated using EC, a filter is used for collection of the aerosol [45]. The analysis method for the EC aerosol used the tobacco analysis method of ISO, WHO, and CORESTA N° 75 [26,30]. To compare the adsorption capacity of TSNAs on each filter media, CFP (44 mm Φ) and QW were prepared [16]. Note that, for this comparative test, a dilution factor in the range of

20 to 100 was chosen to minimize the analytical interference of LC-MS/MS due to the large amount of PG and VG (e.g., >80%) in EC liquid solutions, and to maximize the detection capacity for the TSNAs. The amount of aerosol that is generated per puff and the consumption rate of EC solution were measured by the mass change tracking (MCT) method [46]. The dilution factor can be computed, as follows:

$$\text{Dilution Factor} = \text{Amount of extraction solution (mL)}/\text{Amount of aerosol}$$
$$\text{generated by EC smoke automatic sampling device (mL)}$$

The CFP used as a sampling filter for tobacco mainstream smoke in the ISO, WHO, and CORESTA N° 75 methods has a large diameter (44 mm) and 20 mL of extraction solution is generally required to extract TSNAs that are captured on CFP. In contrast, the QW filter is prepared manually by packing 15 mg of QW into a quartz tube (6.35 mm × 90 mm), and is commonly used for GC-MS–based thermal desorption analysis of indoor air and odor samples [47]. For the evaluation of RR for the CFP, spiked samples were prepared at three concentrations of 500, 1250, and 2500 ng mL$^{-1}$ for the extraction of TSNAs captured on CFP using 10 mL extraction solution (dilution factor: ~330). In the case of the QW filter, relatively low concentration levels of 100, 500, and 1000 ng mL$^{-1}$ of spiked samples were prepared while considering the need for a small amount (1.8 mL) of extraction solution (dilution factor: ~60) as compared to CFP. Spiked samples that were collected using two types of sampling filters (CFP and QW) after solvent extraction were prepared at concentrations of 2, 5, and 10 ng mL$^{-1}$ to allow quantitative recovery analysis using a LC-MS/MS system.

An automatic sampling device for EC smoke designed by Chemtekins Co. (Sungnam, Korea) was used to precisely control the rate of generation of EC aerosol [41]. The EC device filled with either a spiked sample or a commercial E-liquid was placed in the EC smoke automatic sampling device for the production of aerosols. For triplicate analysis, the vaped sample was collected onto three individual filters. Sampling of the aerosol onto each filter of CFP and QW was conducted at a sampling (purging) velocity of 1 L min$^{-1}$, with the following puff conditions: puff interval, 10 s; puff duration, 2 s; and, the number of puffs, 10. The CFP-collected aerosol samples (A-CFP) were extracted with 10 mL of AA and ultrasonicated for 30 min. Extracted solution (2 mL) was then filtered through a syringe filter (0.45 μm, PTFE) and transferred to a 2 mL vial. In the case of sampling vapor onto QW filter (A-QW), 1.5 g (~1.8 mL) of AA was filtered directly through a QW filter and transferred to a 2 mL vial. The aerosol sample collected on the QW filter according to the puff conditions selected in this study (as described above) is approximately 30 mg. Hence, the dilution factor of the extraction solvent is about 50. The extraction solution was transferred to a 2 mL amber vial for convenience of final analysis and a 1-μL sample from the two types of filtered extract solution was analyzed by LC-MS/MS. A schematic diagram of analysis of the aerosol sample for both types of filter is shown in Supplementary Figure S1 online.

## 2.4. Setup of LC-MS/MS System

Spiked and commercial EC samples were analyzed by HPLC (LC-20AD, Shimadzu, Japan)-MS/MS (LCMS-8040, Shimadzu, Kyoto, Japan) (Table 1). After the automatic injection of 1 μL of sample into the HPLC system, the four target compounds were separated using a ZORBAX Eclipse Plus C18 column (film thickness: 0.35 μm, diameter: 3 mm, length: 150 mm; Agilent, St. Louis, MO, USA) with the column temperature of 55 °C. Two mobile phases with a flow rate of 0.4 mL min$^{-1}$ were used to elute the analyte: solvent A (0.1% acetic acid in deionized water [DW]) and solvent B (0.1% acetic acid in methanol [MeOH]). A mobile gradient with 55% solvent B was shown in Table 1(A).

The MS/MS system was operated with electrospray ionization in the positive ion mode (ESI+). Ion spray voltage and ion spray temperature were set to 4.5 kV and 700 °C, respectively. Pure nitrogen gas was used as a nebulizing gas at a flow rate of 3 L min$^{-1}$ to make fine droplets. A drying gas flow rate of 15 L min$^{-1}$, which desolvated the droplet, was employed. Argon gas was used as a collision gas

that fragmented the ions at a pressure of 230 kPa. Three ion transition pairs were then detected in the multiple reaction monitoring (MRM) mode. These pairs were presented (precursor ion to three product ions), the collision energy and dwell time for the four TSNA compounds are shown in Table 1(B,C).

**Table 1.** Operational condition of high-performance liquid chromatography (HPLC)–tandem mass spectrometry (MS/MS) system and multiple reaction monitoring (MRM) conditions for the analysis of tobacco-specific nitrosamines (TSNAs).

| (A) High-Performance Liquid Chromatography (LC-20AD, Shimadzu, Japan) | | | | | | | |
|---|---|---|---|---|---|---|---|
| **Column: ZORBAX Eclipse Plus C18 (Agilent, USA), 3 × 150 mm, Particle size: 3.5 μm** | | | | | | | |
| Oven temp: | 55 | | | | °C | | |
| Injection volume: | 1 | | | | μL | | |
| Flow rate: | 0.4 | | | | mL min$^{-1}$ | | |
| Pump mode: | Binary gradient | | | | | | |
| Mobile phase A: | 0.1% Acetic acid in water | | | | | | |
| Mobile phase B: | 0.1% Acetic acid in methanol | | | | | | |
| Gradient: | Time (min) | 0 | 4 | 7 | 8 | 20 | 25 |
| | Solvent B (%) | 55 | 98 | 98 | 2 | 2 | 55 |
| Total analysis time: | 25 | | | | min | | |

| (B) Tandem Mass Spectrometry (LCMS-8040, Shimadzu, Japan) | | |
|---|---|---|
| Acquisition mode: | MRM | |
| Electrospray ionization: | ESI | mode |
| Polarity: | Positive | |
| Nebulizing gas flow (N$_2$): | 3.0 | mL min$^{-1}$ |
| Drying gas (N$_2$): | 15.0 | mL min$^{-1}$ |
| CID gas (Ar): | 230 | kPa |
| Interface voltage: | 4.5 | kV |
| Loop time: | 0.348 | s |
| Dwell time: | 26 | msec |

**(C) Mass spectrum parameters for MRM condition**

| Compounds | Precursor Ion (*m/z*) | Product Ion | | CE * (V) | Dwell time (m s) |
|---|---|---|---|---|---|
| | | Quantifier (*m/z*) | Qualifier (*m/z*) | | |
| NNN | 178 | 148.1 | | 12 | 26 |
| | | | 120.1 | 23 | 26 |
| | | | 119.1 | 33 | 26 |
| NNK | 208 | 122.1 | | 11 | 26 |
| | | | 79.1 | 30 | 26 |
| | | | 106.1 | 18 | 26 |
| NAT | 190 | 160 | | 13 | 26 |
| | | | 79.1 | 39 | 26 |
| | | | 106.1 | 22 | 26 |

* Collision Energy.

### 2.5. Purchase of Commercial EC Solution

The six types of EC liquid solution that were tested in this study were purchased at a retail store in August 2017. The criteria for selecting the flavor or contents of nicotine were chosen randomly. Supplementary Table S3 shows the basic information of the commercial EC solutions, except for manufacturer name.

## 3. Results and Discussion

### 3.1. Calibration and Quality Assurance/Quality Control

Table 2 shows the calibration results and summary of the quality assurance/quality control (QA/QC) information of L-WS using a LC-MS/MS system. Figure 1 shows the retention times for individual TSNA compounds under these instrumental conditions. The standard calibration curve was examined by computing a regression line of peak area ratio for weight (pg) of TSNAs in L-WS using the least squares method. All of the calibration curves of four TSNA compounds showed

excellent linearity ($R^2 > 0.999$) in the concentration range of 0.5 to 100 ng mL$^{-1}$ (Figure S2). The MDL for TSNA was calculated as the product of the standard deviation of seven replicate analyses based on the Student's t-value (t = 3.14 for degrees of freedom = 6) at 99% confidence level according to US EPA guidelines. The MDL of NNN, NNK, NAT, and NAB were estimated as 4.40, 4.47, 3.71, and 3.28 ng mL$^{-1}$, respectively. The relative standard error (RSE) was stable at ~2% at a L-WS concentration of 5 ng mL$^{-1}$.

**Table 2.** Results of calibration analysis and quality assurance (QA) information obtained using liquid phase working standards (L-WS).

| Order | Compound | RF [a] | $R^2$ | RSE (%) [b] | MDL (pg) [c] | MDL (ng mL$^{-1}$) [c] |
|---|---|---|---|---|---|---|
| 1 | NNN | 40,707 | 0.9999 | 0.74 | 4.40 | 4.40 |
| 2 | NNK | 79,308 | 0.9999 | 0.49 | 4.47 | 4.47 |
| 3 | NAT | 82,614 | 0.9999 | 1.17 | 3.71 | 3.71 |
| 4 | NAB | 53,392 | 0.9997 | 0.59 | 3.28 | 3.28 |

[a] Response factor value = (Peak area of each compound)/(Injected mass amount); [b] Relative standard error (5 ng mL$^{-1}$ of L-WS was used to measure RSE values) = (Standard deviation/Mean)/$\sqrt{3} \times 100 = CV/\sqrt{3}$; [c] Method detection limit.

**(a)**

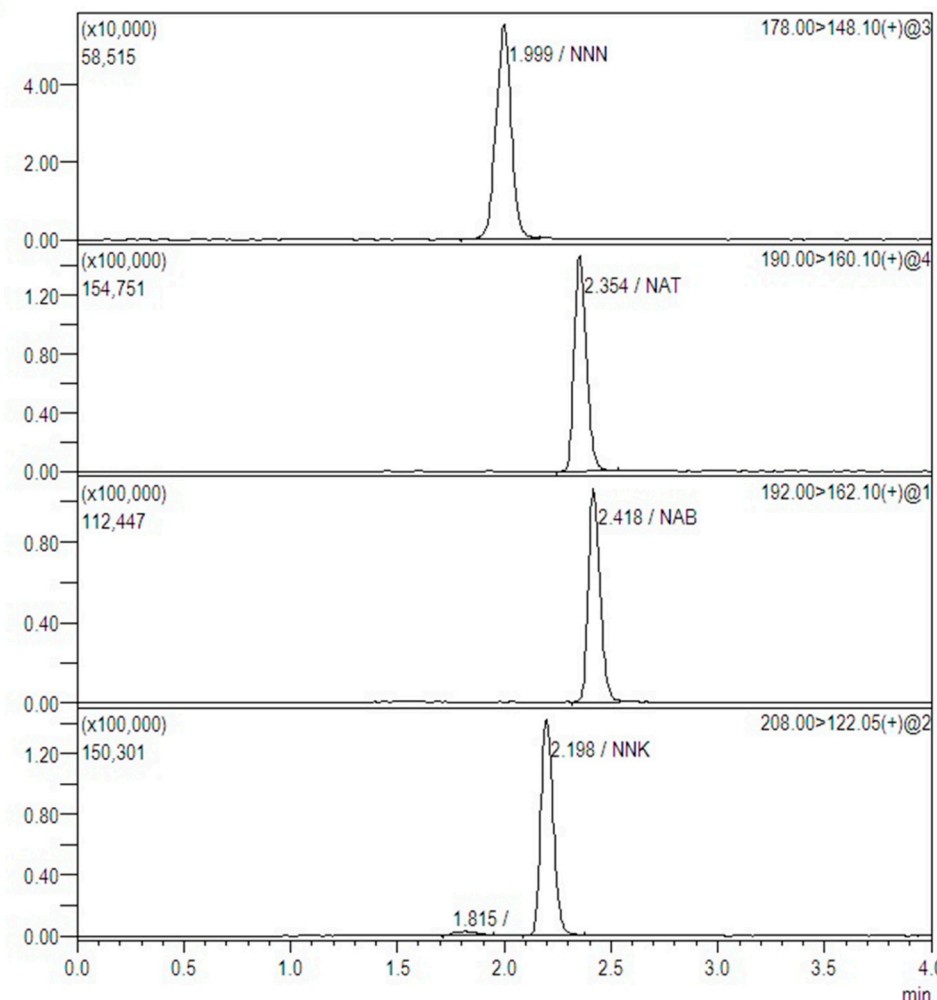

**Figure 1.** *Cont.*

**(b)**

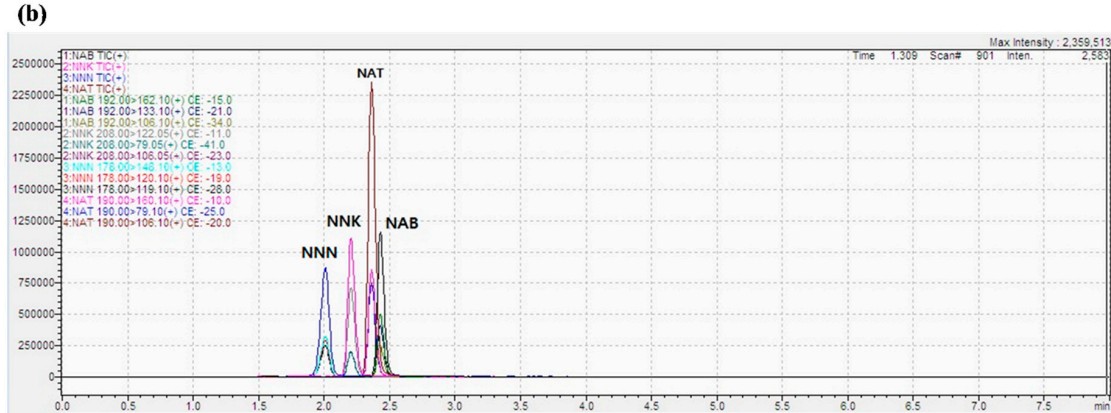

**Figure 1.** Chromatograms of TSNA for the 10 ng mL$^{-1}$ of working standards by liquid chromatography-tandem mass spectrometry (LC-MS/MS) (**a**) extracted ion chromatogram (EIC) of quantifier ions for each target compound and (**b**) Chromatogram in Total ion current (TIC) mode with both quantifier and qualifier ions.

### 3.2. Determination of Relative Recovery for Dilution Solutions

To compare the dilution capacity according to the dilution factor, the RRs of four TSNA compounds that were computed by the response factor (RF) of L-WS are shown in Supplementary Figure S3 online. Comprehensive examination of two types of RR test showed that the RRs of ACN were considerably higher than those of AA. In the L-S1 phase, assessment of the average RR value of all four TSNAs for each dilution solution showed a high recovery of 137 ± 16.6% for ACN as compared with 109 ± 15.6 for AA. The RRs of four TSNAs in a 20-fold diluted sample with AA were considerably high (135 ± 0.53 (NNN), 113 ± 0.12 (NNK), 108 ± 0.47 (NAT), and 136 ± 0.17% (NAB); Table 3). In addition, unidentified peaks appeared due to contamination of the C18 column, which may be caused by the high contents of PG and VG in the EC liquid solution. When the sample was diluted 50-fold, the RRs of NNK and NAT were stable, while NNN and NAB exhibited relatively high recoveries (114 ± 0.24 for NNN, 95.7 ± 0.01 for NNK, 93.5 ± 0.16 for NAT, and 113 ± 0.13% for NAB). In contrast, the RRs of three TSNAs for the 100-fold diluted sample were stable with the exception of NNN (115 ± 0.20 for NNN, 88.7 ± 0.12 for NNK, 91.3 ± 0.01 for NAT, and 104 ± 0.07% for NAB). According to a previous study, the maximum concentration of TNSA detected in one of the commercial products was 60.8 ng mL$^{-1}$ for NNN [48]. If the E-solution is diluted 100-fold with AA, analysis of the low concentration sample is unreliable because the concentration of the diluted sample may fall outside the calibration range. In contrast, the RRs of all samples diluted with ACN were greater than 110%, regardless of the dilution factor. The 100-fold diluted sample showed the lowest RR across the three dilutions (20, 50, and 100 fold), but it was still much higher than the RR of the 100-fold diluted sample with AA (difference up to 42%). Therefore, AA was better than ACN for the dilution of the EC solution with respect to stable recovery.

In the L-S2 phase, the dilution factor was fixed at 50 based on the results of experiment L-S1. The RRs of four TSNAs in spiked samples with two types of dilution solution are compared in Table 4. When diluting the samples with ACN, the RR was approximately 100% for all TSNAs at a low concentration (50 ng mL$^{-1}$) of spiked samples (98.8 ± 3.11 for NNN, 102.1 ± 11.4 for NNK, 91.2 ± 8.06 for NAT, and 99.3 ± 8.77% for NAB), but increased drastically at the highest concentration of 500 ng mL$^{-1}$ (273 ± 9.52 for NNN, 332 ± 30.7 for NNK, 251 ± 20.8 for NAT, and 261 ± 22.0% for NAB). It is hard to explain this high recovery by only the matrix effect. In contrast, the average RRs of four TSNAs in all samples that were diluted with AA were stable at all concentrations: 83.1 ± 4.65 for NNN, 88.0 ± 5.16 for NNK, 81.5 ± 3.63 for NAT, and 88.8 ± 4.45% for NAB. Therefore, a 50-fold dilution of the E-liquid with AA was determined as a preferable pretreatment option for analysis of TSNAs in EC solution.

**Table 3.** Relative recoveries of four target TSNAs in relation to the type of extraction solution (ammonium acetate (AA) and acetonitrile (ACN)) and dilution factor (20, 50, and 100 fold) using one point spiking of TSNA in L-S1 solution.

| Order | Sample Code [a] | Compounds | | | |
|---|---|---|---|---|---|
| | | NNN | NNK | NAT | NAB |
| (A) Theoretical concentration of TSNA in the spiked sample diluted with two solutions (ng mL$^{-1}$) [b][c] | | | | | |
| 1 | Spiked sample | 103 | 103 | 101 | 103 |
| (B) Detected concentration of TSNA in the spiked sample by LC-MS/MS (ng mL$^{-1}$) [b] | | | | | |
| 1 | L-AA-100 | 1.19 | 0.92 | 0.93 | 1.07 |
| 2 | L-AA-50 | 2.35 | 1.98 | 1.89 | 2.33 |
| 3 | L-AA-20 | 7.00 | 5.83 | 5.46 | 7.04 |
| 4 | L-ACN-100 | 1.63 | 1.21 | 1.18 | 1.37 |
| 5 | L-ACN-50 | 3.08 | 2.59 | 2.38 | 2.93 |
| 6 | L-ACN-20 | 8.61 | 6.97 | 6.60 | 7.80 |
| (C) Relative recovery, RR (%) [b] | | | | | |
| 1 | L-AA-100 | 115 | 88.7 | 91.3 | 104 |
| 2 | L-AA-50 | 114 | 95.7 | 93.5 | 113 |
| 3 | L-AA-20 | 135 | 113 | 108 | 136 |
| 4 | L-ACN-100 | 158 | 117 | 117 | 133 |
| 5 | L-ACN-50 | 149 | 125 | 118 | 142 |
| 6 | L-ACN-20 | 167 | 135 | 130 | 151 |

[a] Sample code was assigned by analytical condition of EC liquid solution plus type of chemical solutions used for dilution (ammonium acetate (AA) (100 mmol) or acetonitrile (ACN)) and the extent of dilution (i.e., dilution factors: 100, 50, and 20). [b] The values are expressed as mean values. [c] 10 ng mL$^{-1}$ final concentration was excluded because of the high dilution factor.

**Table 4.** Relative recoveries of four target TSNAs in relation to the type of extraction solution (AA and CAN) and concentration levels of spiking (50, 250, and 500 ng mL$^{-1}$) in L-S2 solution: Here, the dilution factor was fixed at 50 fold.

| Order | Sample Code [a] | Compounds | | | |
|---|---|---|---|---|---|
| | | NNN | NNK | NAT | NAB |
| (A) Theoretical concentration of TSNA in the spiked sample (ng mL$^{-1}$) (dilution factor: 50) [b] | | | | | |
| 1 | L-AA-1 | 53.0 | 53.0 | 51.9 | 53.0 |
| 2 | L-AA-5 | 256 | 256 | 251 | 256 |
| 3 | L-AA-10 | 516 | 516 | 505 | 516 |
| 4 | L-ACN-1 | 53.0 | 53.0 | 51.9 | 53.0 |
| 5 | L-ACN-5 | 256 | 256 | 251 | 256 |
| 6 | L-ACN-10 | 516 | 516 | 505 | 516 |
| (B) Detected concentration of TSNA in the spiked sample by LC-MS/MS by LC-MS/MS (ng mL$^{-1}$) [b] | | | | | |
| 1 | L-AA-1 | 43.4 | 48.4 | 45.0 | 45.2 |
| 2 | L-AA-5 | 226 | 232 | 206 | 241 |
| 3 | L-AA-10 | 408 | 424 | 393 | 450 |
| 4 | L-ACN-1 | 52.4 | 54.1 | 48.3 | 52.6 |
| 5 | L-ACN-5 | 246 | 270 | 233 | 254 |
| 6 | L-ACN-10 | 1410 | 1715 | 1270 | 1352 |
| (C) Relative recovery, RR (%) [b] | | | | | |
| 1 | L-AA-1 | 82.0 | 91.4 | 84.9 | 85.3 |
| 2 | L-AA-5 | 88.2 | 90.6 | 82.0 | 93.8 |
| 3 | L-AA-10 | 79.1 | 82.1 | 77.7 | 87.3 |
| 4 | L-ACN-1 | 98.8 | 102 | 91.2 | 99.3 |
| 5 | L-ACN-5 | 95.8 | 105 | 92.9 | 99.0 |
| 6 | L-ACN-10 | 273 | 332 | 251 | 262 |

[a] Sample code was assigned by analytical condition of EC liquid solution with dilution solutions (ammonium acetate (AA) (100 mmol) or acetonitrile (ACN)) and concentrations of diluted samples (1.00, 5.00, and 10.0 ng mL$^{-1}$). [b] The values in Table are expressed as mean values.

### 3.3. Determination of Relative Recovery for Sampling Filter

The RRs of four TSNAs in aerosol captured on the filters were measured by the RF of L-WS and MCT approach (Supplementary Table S1) [41,46,48]. The average RRs of four TSNAs that were collected onto CFP were relatively low: 78.0 ± 2.04 for NNN, 61.3 ± 3.36 for NNK, 50.2 ± 2.05 for NAT, and 50.8 ± 1.74% for NAB. The low recovery of CFP might be attributed to the high dilution factor (365 ± 40.2, Supplementary Table S2) in the extraction procedure. In the process of puffing with the automatic sampling device a small amount of smoke did not pass through the filter and leaked out to the EC air hole.

In all of the aerosol samples that were collected on the QW filter, NNN and NAB had high recovery relative to NNK and NAT. The RRs of four TSNAs for each concentration of spiked sample were more stable at low concentrations (92.2 ± 6.98 for NNN, 82.4 ± 4.00 for NNK, 82.7 ± 9.40 for NAT, and 101 ± 8.07% for NAB, in 100 ng mL$^{-1}$) as compared to high concentrations (77.5 ± 12.1 for NNN, 66.2 ± 2.00 for NNK, 67.7 ± 9.30 for NAT, and 88.5 ± 10.2% for NAB, in 1000 ng mL$^{-1}$). The concentrations that were calculated after pretreatment for each filter were 54.5 ± 11.0 for CFP and 89.6 ± 8.71% for QW. Therefore, high recovery and low relative standard error (RSE) values were observed for the QW filter (Figure 2).

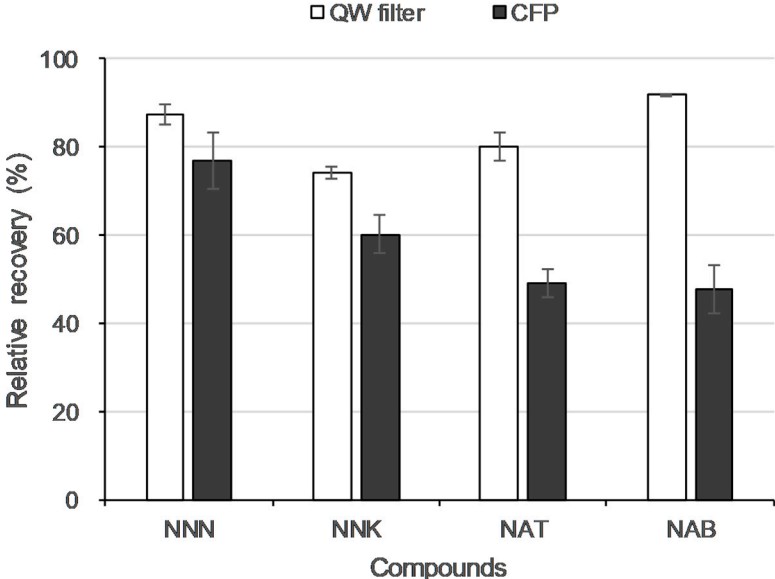

**Figure 2.** Comparison of relative recovery (RR) for the four TSNAs between quartz wool (QW) filter and Cambridge filter pad (CFP). RR (%) = (Concentration of spiked sample detected by LC-MS/MS system)/(Theoretical concentration of spiked sample) × 100.

### 3.4. Evaluation of Commercial EC Product

Six commercial EC solution products that were purchased randomly were analyzed using the method of liquid and aerosol samples after spiking (Supplementary Table S3). None of our target TSNAs were detected in the analysis of commercial EC solutions. This is probably because of the low content of nicotine in the commercial EC products. In previous studies, the amount of TSNAs was highly correlated with the nicotine and nitrite content in the E-solution cartridge. The amount of the four TSNAs in the EC solution is extremely small, which is consistent with the low amount of nicotine in the drug product: 8.18 ng g$^{-1}$ of four total TSNAs on 16 mg of nicotine in cartridge [35,49,50]. This implies that the six types of EC solution that were tested in this study probably rely on using a small amount of nicotine with the impurities removed. It is therefore necessary to examine the connection between TSNAs and nicotine by analyzing EC solutions with high nicotine content.

Some TSNAs in cigarette mainstream are known to be synthesized during smoking [51]. However, it should be noted that there is no study on the formation of TSNAs in the process of aerosolization from

EC solution. Total TSNA and nitrate levels in 1 mL of EC solution were at least 400 and 1300 times lower than the contents of TSNAs and nitrate per gram of common cigarette, respectively [52]. The amount of total TSNAs in EC vapor when puffing 99 times was at least 27 times lower than the quantity in mainstream per cigarette [53]. In addition, the yield of nicotine that was vaped from EC was 85% lower than that of traditional cigarettes [53], supporting the notion that the low concentration of TSNAs in the aerosol is similar to or even less than the TSNA level in E-liquid. Based on comparison of the EC solution and aerosol, TSNAs did not form during the generation of aerosol in the heating coil.

With EC, it is possible to increase the number of puffs as the EC can be refilled with E-solution, unlike tobacco with a limited number of puffs. According to a survey, EC users inhaled more than four times more aerosols than smokers for 10 min and had an average of $225 \pm 59$ puffs per day [54,55]. Therefore, further experiments will be needed to understand the different behavior between EC liquid and aerosol based on the quantitation of TSNAs present in the aerosol of EC with consideration of the number of puffs.

## 4. Conclusions

In this study, an analysis method for TSNAs in the liquid and aerosol of EC was developed and validated using LC-MS/MS. The liquid phase samples were analyzed after diluting the EC solution under two variable conditions: (1) dilution factor and (2) dilution solution. Dilution of liquid phase samples using two different solvents, AA (100 mmol) and ACN, was assessed in terms of RR. This represents a balance between diluting the sample enough so it is compatible with the analytical system, but not diluting it so much that the detection limits are compromised. The EC solution that was diluted 50-fold with AA showed stable RRs ($83.1 \pm 4.65$ for NNN, $88.0 \pm 5.16$ for NNK, $81.5 \pm 3.63$ for NAT, and $88.8 \pm 4.45\%$ for NAB). The gaseous phase samples were analyzed by extracting the filter captured EC aerosol with the solvent (AA) through the MCT approach. The sampling method of the EC aerosol sample using two different sampling filters (CFP and QW) was also assessed in terms of RR. The QW filter captured EC vapor efficiently and it showed stable RR of $84.3 \pm 13.7$ for NNN, $73.3 \pm 10.9$ for NNK, $75.3 \pm 16.3$ for NAT, and $93.8 \pm 12.3\%$ for NAB. Also, the QW filter has the advantage of a comparatively low dilution factor and consumption of a small amount of extraction solvent (AA). Under the estimated analysis conditions, the MDL values of NNN, NNK, NAT, and NAT were 4.40, 4.47, 3.71, and 3.28 ng mL$^{-1}$, respectively. In tests of six commercial ECs, we detected none of the four TSNAs in E-solution and aerosol. It is well known that a portion of the TSNAs is related to the content of nitrogen oxides. Based on the theoretical relationship of TSNAs and nitrogen oxide, the six EC solutions have very small amounts of TSNAs that are undetectable. A follow-up study should be performed to quantify TSNAs according to the nitrogen content to confirm the generation or transfer characteristics of TSNAs.

**Supplementary Materials:** The following are available online at http://www.mdpi.com/2076-3417/8/12/2699/s1, this manuscript accompanies supplementary information.

**Author Contributions:** Y.-S.L., K.-H.K., S.S.L., and R.J.C.B. contributed toward the preparation of the manuscript text, figures, and tables. Y.-S.L. and S.-H.J. conducted all major experiments.

**Funding:** This research received funding as described in Acknowledgments.

**Acknowledgments:** This study was supported by a grant from the National Research Foundation of Korea (NRF) funded by the Ministry of Science, ICT, & Future Planning (No. 2016R1E1A1A01940995). This research was supported by a grant (14182MFDS977) from the Ministry of Food and Drug Safety in 2017.

**Conflicts of Interest:** The authors declare no conflict of interests in financial/non-financial or any kinds of interests.

**Data Availability:** The datasets generated from this study are included in this published article and in Supplementary information.

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
