# Peer review of "Analytical Method for Measurement of Tobacco-Specific Nitrosamines in E-Cigarette Liquid and Aerosol"

_applsci, doi:10.3390/app8122699_

Round 1

Reviewer 1 Report

The authors present an interesting method for measurement of tobacco-specific nitrosamines in E-cigarette liquid and aerosol. The authors have designed appropriate all the protocols and have reported clearly the results and conclusions of their study. In addition, the authors present the most of the criteria for validation study.

More detailed comments:

The authors should identify how they calculated the limit of detection of the method.

Also, please make more clear the reproducibility data: e.g. add to the corresponding tables the calculated coefficient of variation %CV.

Table 2 (A) is confusing. The data in A and B are not in good agreement. Please correct. Table 2 (B) is ok.

It is also preferred to add all the calibration graphs in the supplementary material.

As the analysis of commercial EC solution products did not revealed any of the target TSNAs studied, I suggest that the authors should perform some extra recovery studied using these EC solutions. In addition, as the authors suggest, they should also try samples with increased concentrations of nicotine to present a more complete study.

Author Response

Rev 1

[1] The authors present an interesting method for measurement of tobacco-specific nitrosamines in E-cigarette liquid and aerosol. The authors have designed appropriate all the protocols and have reported clearly the results and conclusions of their study. In addition, the authors present the most of the criteria for validation study.

Ans] Thank you for your interest in our paper. The comments you have made are reflected in this paper and highlighted in yellow.

More detailed comments:

[2] The authors should identify how they calculated the limit of detection of the method.

Ans] We added how to compute the method detection limit (MDL) in the part 3.1. (Page 6)

[3] Also, please make more clear the reproducibility data: e.g. add to the corresponding tables the calculated coefficient of variation %CV.

Ans] The equations are given as follows and explained in Table 2. CV (%) = Standard deviation / Mean * 100. RSE (%) = (Standard deviation / Mean)/ . Hence, RSE can be expressed as CV/ (In the root, the number of triplicate analyses (n=3)).

[4] Table 2 (A) is confusing. The data in A and B are not in good agreement. Please correct. Table 2 (B) is ok.

Ans] First, we modified the table number in order (Table 2 -> Table 3).

The data in A was agreement with B.  The detailed results in Table 3-A were explained in part 3.2 (1st paragraph in Page 7). The result in Table 3-B was in the part 3.2 2nd paragraph in Page 7.

[5] It is also preferred to add all the calibration graphs in the supplementary material.

Ans] We added Figure 2. Calibration curves of TSNA are now in the supplementary material. Thank you.

Reviewer 2 Report

Lee et al, have carried out a method optimization and then quantified the levels of 4 tobacco specific nitrosamines from E-cigarette solutions. This work is very interesting and relevant for today society with many people using these systems, with some taking up E-cigarettes having never smoked in some instances. This work will be of wide interest to public health and analytical chemistry researchers. There are several points which require attention in the manuscript prior to being ready for publication.

Major comments:

There doesn’t appear to be any internal standards used making it impossible to quantify any results

There is also no method validation in terms of retention time and peak area reproducibility, these are required for method development.

Introduction

Page 3 line 73: What is meant by detectability? Should this be either sensitivity or the ability to ionize some compounds with EI compared to ESI?

Page 3 line 76: Please correct to Food and Drug Administration

Page 3 line 82: Is there any information to support this assumption in the literature already?

Page 3 line 84: Change filer sampling to filter sampling

Page 3 line 87: The acronyms for ISO, WHO and CORESTA N°75 need to be defined

Materials and methods

Page 4 line 121: “the analysis system can be damaged if….” This is poorly phrased and should instead state that high concentrations of PG and VG ….. are incompatible with LC-MS/MS.

Page 5 line 131: Remove the word automatically, this is used throughout the manuscript to describe automatic analysis and should be removed in all cases

Page 5 line 140: Can more detail of how the filter was used be added it’s an important and interesting step for this method development.

Page 6 lines 162-163: Again, can more details of this system be given as this is an important process in the method.

Page 6 line 181: It’s not clear what model the mass spectrometer is could this be clarified

Page 6 line 184: “Two mobile solvents” change to two mobile phases

 Page 6 line 191: Changes Cracked to fragmented

Table A (change to Table 1) In the MS parameters the precursor ion is a nominal mass whereas the quantifier and qualifier are to 1 decimal place. The same number of decimal places should be used throughout.

Figure 1: The bottom half of the figure is missing

Table 2: Why is NAT spiked at a lower concentration?

Author Response

Rev 2

Comments and Suggestions for Authors

[1] Lee et al, have carried out a method optimization and then quantified the levels of 4 tobacco specific nitrosamines from E-cigarette solutions. This work is very interesting and relevant for today society with many people using these systems, with some taking up E-cigarettes having never smoked in some instances. This work will be of wide interest to public health and analytical chemistry researchers. There are several points which require attention in the manuscript prior to being ready for publication.

Ans] Thank you for your interest in our paper. The comments you have made are reflected in this paper and highlighted in yellow

Major comments:

[2] There doesn’t appear to be any internal standards used making it impossible to quantify any results

Ans]  We wanted to present a simple and fast method to detect TSNA without the use of internal standards.

[3] There is also no method validation in terms of retention time and peak area reproducibility, these are required for method development.

Ans] Reproducibility is presented by calculating relative standard error (RSE (%)) in Table 2. Retention time for peak is shown in Figure 1. LC-MS MS can detect only target compounds by scanning paper.

[4] Introduction

Page 3 line 73: What is meant by detectability? Should this be either sensitivity or the ability to ionize some compounds with EI compared to ESI?

Ans] The meaning of detectability include both sensitivity and ability to ionize some compounds with EI compared to ESI. LC-MS/MS may be programmed to select certain ions to fragment. As long as there are no interferences or ion suppression, the LC separation can be made quite quickly. We want to express advantages of LC/MSMS to one word "detectability".

[5] Page 3 line 76: Please correct to Food and Drug Administration

Ans] We corrected to FDA full name, Thank you for suggestion.

[6] Page 3 line 82: Is there any information to support this assumption in the literature already?

Ans] There are various types of nicotine as informed on the bottle of commercial EC solution. Also we add the reference for synthetic nicotine in page 3.

[7] Page 3 line 84: Change filer sampling to filter sampling

Ans] Thank you for letting us know the typo. We modified the word.

[8] Page 3 line 87: The acronyms for ISO, WHO and CORESTA N°75 need to be defined

Ans] We defined the acronyms for ISO, WHO, and CORESTA. Thank you.

[9] Materials and methods

Page 4 line 121: “the analysis system can be damaged if….” This is poorly phrased and should instead state that high concentrations of PG and VG ….. are incompatible with LC-MS/MS.

Ans] Thank you for kindly advice. When we look at the dictionary definition of a word "incompatible", it is different from our intention. So I modified the sentence in Page 4.

[10] Page 5 line 131: Remove the word automatically, this is used throughout the manuscript to describe automatic analysis and should be removed in all cases

Ans] Thanks for your suggestion. I removed the word.

[11] Page 5 line 140: Can more detail of how the filter was used be added it’s an important and interesting step for this method development.

Ans] We explained how to collect aerosol using the 2 types of filters in page 5 and showed in Figure S1.

[12] Page 6 lines 162-163: Again, can more details of this system be given as this is an important process in the method.

Ans] Details of LC-MS/MS system are described in part 2.4 in Page 6.

[13] Page 6 line 181: It’s not clear what model the mass spectrometer is could this be clarified

Ans] We provided the model of mass spectrometer (LCMS-8040, Shimadzu, Japan) in Page 6.

[14] Page 6 line 184: “Two mobile solvents” change to two mobile phases

Ans] We changed the word. Thanks for your suggestion.

[15] Page 6 line 191: Changes Cracked to fragmented

Ans] We changed the word. Thanks for your suggestion.

[16] Table A (change to Table 1) In the MS parameters the precursor ion is a nominal mass whereas the quantifier and qualifier are to 1 decimal place. The same number of decimal places should be used throughout.

Ans] First, we changed the table number in order. We changed the decimal place.

[17] Figure 1: The bottom half of the figure is missing

Ans] We inserted the same figure again.

[18] Table 2: Why is NAT spiked at a lower concentration?

Ans] We made the TSNA liquid working standard using the 1 mg/mL ample of NNN, NNK, NAB, and 5 mg of NAT powder in Sigma-aldrich. So concentration of NAT in TSNA mixture was slightly different.

[19] As the analysis of commercial EC solution products did not revealed any of the target TSNAs studied, I suggest that the authors should perform some extra recovery studied using these EC solutions. In addition, as the authors suggest, they should also try samples with increased concentrations of nicotine to present a more complete study.

Ans] We have the plan of the quantification of TSNA in commercial EC solution in the future paper.

Reviewer 3 Report

The Author developed and validated a method for the collection and analysis of tobacco specific nitrosamines (TSNAs) that are present in electronic cigarette (EC) liquid or released from aerosol samples using a liquid chromatography-tandem mass spectrometry. That appears as the main concern of the present paper. The paper is well written and the experimental design is consistent with the conclusion. The author clearly declare that is well known that a portion of the TSNAs is related to the content of NOx. Moreover considering that they have focused their attention on the analytical methods of target compounds I formally retain that before the publication  the Authors have to consider previous published papers like Cancer Epidemiol Biomarkers Prev 2008;17(12). December 2008 and Int. J. Environ. Res. Public Health 2015, 12, 9046-9053; and other several papers about the target compounds that they are evaluating. This in order to improve their discussion about the possible follow-up study about the quantification of TSNAs according to the nicotine content and confirm the generation or transfer  characteristics of TSNAs. Human health and safety of the possible consumers have to be consider before than the analytical method to increase the audience of the possible readers.

Several points:

Please add a space after the number and °C notation

Figure 2 report a no complete picture of the chromatogram of standard solution probably it may be more convenient shows a chromatogram of the extraction 

Author Response

Rev 3

[1] The Author developed and validated a method for the collection and analysis of tobacco specific nitrosamines (TSNAs) that are present in electronic cigarette (EC) liquid or released from aerosol samples using a liquid chromatography-tandem mass spectrometry. That appears as the main concern of the present paper. The paper is well written and the experimental design is consistent with the conclusion. The author clearly declare that is well known that a portion of the TSNAs is related to the content of NOx. Moreover, considering that they have focused their attention on the analytical methods of target compounds I formally retain that before the publication the Authors have to consider previous published papers like Cancer Epidemiol Biomarkers Prev 2008;17(12). December 2008 and Int. J. Environ. Res. Public Health 2015, 12, 9046-9053; and other several papers about the target compounds that they are evaluating. This in order to improve their discussion about the possible follow-up study about the quantification of TSNAs according to the nicotine content and confirm the generation or transfer characteristics of TSNAs. Human health and safety of the possible consumers have to be consider before than the analytical method to increase the audience of the possible readers.

Ans] Thank you for showing your interest in our paper. We added the reference you recommended. The comments you have made are reflected in this paper and highlighted in yellow.

 Several points:

[1] Please add a space after the number and °C notation

Ans] Thank you for suggestion. I added the space after the number and ℃ notation.

[2] Figure 2 report a no complete picture of the chromatogram of standard solution probably it may be more convenient shows a chromatogram of the extraction.

Ans] Thank you for your suggestion. We added the chromatogram of extraction for TSNA in TIC mode (Figure 1-(b)).

Round 2

Reviewer 1 Report

The authors confronted well with reviewers' suggestions.

Author Response

Thanks for nice review.

Reviewer 2 Report

Comments replied to sufficiently to resolve issues

Author Response

Thanks for nice review.

Reviewer 3 Report

The paper has been improved as previosly requested. 

Author Response

Thanks. Now suggested refs are all added in the revised ms.